# Using experimental gaming simulations to elicit risk mitigation behavioral strategies for agricultural disease management

Eric M. Clark[1,2]*, Scott C. Merrill[1,2,3], Luke Trinity[1,4], Gabriela Bucini[1,2], Nicholas Cheney[1,5], Ollin Langle-Chimal[1,4], Trisha Shrum[1,6], Christopher Koliba[1,3,6], Asim Zia[1,3,6], Julia M. Smith[1,7]

1 SEGS Lab, University of Vermont, Burlington, Vermont, United States of America, 2 Department of Plant and Soil Science, University of Vermont, Burlington, Vermont, United States of America, 3 Gund Institute for Environment, University of Vermont, Burlington, Vermont, United States of America, 4 Complex Systems Center, University of Vermont, Burlington, Vermont, United States of America, 5 Department of Computer Science, University of Vermont, Burlington, Vermont, United States of America, 6 Department of Community Development and Applied Economics, University of Vermont, Burlington, Vermont, United States of America, 7 Department of Animal and Veterinary Sciences, University of Vermont, Burlington, Vermont, United States of America

* eclark@uvm.edu

**Data Availability Statement:** All relevant data are within the manuscript and its Supporting Information files. These files will be made available upon acceptance of this manuscript.

## Abstract

Failing to mitigate propagation of disease spread can result in dire economic consequences for agricultural networks. Pathogens like Porcine Epidemic Diarrhea virus, can quickly spread among producers. Biosecurity is designed to prevent infection transmission. When considering biosecurity investments, management must balance the cost of protection versus the consequences of contracting an infection. Thus, an examination of the decision making processes associated with investment in biosecurity is important for enhancing system wide biosecurity. Data gathered from experimental gaming simulations can provide insights into behavioral strategies and inform the development of decision support systems. We created an online digital experiment to simulate outbreak scenarios among swine production supply chains, where participants were tasked with making biosecurity investment decisions. In Experiment One, we quantified the risk associated with each participant's decisions and delineated three dominant categories of risk attitudes: risk averse, risk tolerant, and opportunistic. Each risk class exhibited unique approaches in reaction to risk and disease information. We also tested how information uncertainty affects risk aversion, by varying the amount of visibility of the infection as well as the amount of biosecurity implemented across the system. We found evidence that more visibility in the number of infected sites increases risk averse behaviors, while more visibility in the amount of neighboring biosecurity increased risk taking behaviors. In Experiment Two, we were surprised to find no evidence for differences in behavior of livestock specialists compared to Amazon Mechanical Turk participants. Our findings provide support for using experimental gaming simulations to study how risk communication affects behavior, which can provide insights towards more effective messaging strategies.

**Funding:** This work is/was supported by the USDA National Institute of Food and Agriculture, under award number 2015-69004-23273. The contents are solely the responsibility of the authors and do not necessarily represent the official views of the USDA or NIFA.

**Competing interests:** The authors have declared that no competing interests exist.

## Introduction

Circumventing exposure to an outbreak can require economic foresight and ambition. For managers of agricultural production facilities, biosecurity investment decisions can be challenging, as biosecurity practices are costly and the return on investment can be difficult to quantify (e.g., efficacy uncertainty). At what perceived levels of risk will individuals invest resources to mitigate their chances of becoming infected? For many, this decision-making process is multi-factored and dynamic, where many factors can change over time depending on past experiences [1, 2]. In [3], the psychosocial factors motivating biosecurity adoption with respect to risk attitudes, adherence, and resistance to behavioral change initiatives was shown to be an urgent topic for continued investigation and policy consideration. Studying risk mitigation behavioral strategies can help us understand how information and perceived risk affects the decision-making process.

Livestock epidemics can cause substantial economic damage to agricultural industries, [4] estimates a net annual cost of $900 million to $1.8 billion from PEDv outbreaks. Particularly transmissible pathogens, like Porcine Epidemic Diarrhea virus (PEDv) can spread quickly throughout supply chain networks resulting in high mortality rates [5, 6] and can often reoccur even after an outbreak has been seemingly eradicated [7]. Failure to mitigate an outbreak as well as corresponding "reemergence events" can have serious fiscal consequences [5].

Biosecurity [8] can be defined as a set of tools, practices, disease prevention measures and sanitary regulations designed to attenuate the spread of disease. For example, truck washes have been shown to slow the spread of PEDv [9]. After animal feed was identified as a vector for PEDv [10, 11] sanitation of feed using thermal or chemical treatments helps prevent contamination [12]. In [13], the efficacy of varying degrees of biosecurity for mitigating disease transmission was tested by comparing "Low", "Medium" and "High" biosecurity experimental groups, delineated by the amount of practices implemented. "Medium" and "High" levels of biosecurity were shown to outperform "Low" biosecurity treatments in attenuating disease spread.

In realistic animal supply chain network conditions, investment in biosecurity cannot guarantee safety from an outbreak. Although biosecurity has become an expected norm in crop and animal based agriculture, full participation in these practices is not entirely widespread [3, 14]. Yet, increased biosecurity reduces the likelihood of disease transmission [13]. Each producer's risk of infection is also largely dependent upon their network [15, 16, 17, 18]. Hence, a structural equilibrium exists for optimizing welfare by investing in biosecurity for outbreak mitigation. Our simulations focus on studying participant's aversion to risk in response to perceived economic danger associated with infectious diseases. When there is a disease outbreak, one natural strategy is to "wait and see" how the disease spreads before choosing to allocate resources for protection. These scenarios have been studied with respect to flu vaccines [16, 19]. "Wait and See" strategies were shown to exacerbate outbreaks if the vaccination rate among the population was low. Risk averse individuals, who might vaccinate early, could be perceived as protective shields for those going unvaccinated who are acting as free-riders. This *opportunistic* strategy weighs the perceived opportunity costs of vaccination versus the risk of infection. This behavior may benefit one's own facility, but can increase the chances for an epidemic along with more extreme economic consequences across the supply chain [20].

Computational social science focuses on leveraging data to investigate questions regarding human decision making, behaviors, societies and systems [21–25]. Many have examined decision-making using a behavioral economic lens. For example, the role of risk preferences in decision-making has been widely studied using survey methods such as multiple price lotteries [26, 27]. One method for collecting decision-making data is through the use of serious games.

Experimental gaming simulations that use performance-based incentives can increase salience and effort in the decision making process [28, 29]. Real monetary payments that scale with the payouts in the experimental choices provide an incentive for participants to act according to their true risk preferences. Incentive-compatible experiments are the gold standard for understanding real-world relevant behavior [30, 31]. These experiments can be hosted online to gather social interactions and behaviors from wide audiences [32, 33].

Experimental gaming simulations have also been used to assess the efficacy of digital decision support systems. In regards to emergency response, experimental simulations have been shown to bolster preparedness and reduce economic damage [34]. Interactive simulations have been useful for studying human behavior in the face of an epidemic [35, 36]. Gaming simulations have also been applied for business management decision assessment [37] as well as conceptual training of entrepreneurial strategy [38].

In our experimental process, we intend to extract and compare behavioral strategies that emerge over the course of the simulation. We developed a *biosecurity adoption* metric, based upon players' decisions to allocate resources to reduce their risk of infection. Here we define behavioral scores using the infection rate as our experimental variable. We calculated each participant's biosecurity adoption rating by tallying their level of protection (None, Low, Medium, or High) across each simulated year, consisting of 6 "decision months". Decisions to adopt biosecurity earlier in the year were implicitly weighted heavier than investing at the end of the year. Participants were informed of the probability of infection across several high risk and low risk scenarios, in which we also varied the amount of biosecurity in their simulated network. This allowed us to compare each player's risk aversion score for both low and high probabilities of infection as a way of measuring their behavioral response with respect to their perceived risk.

Aggregated biosecurity adoption ratings were then categorized using clustering algorithms. Clustering algorithms [39] are unsupervised learning methods for grouping multi-dimensional data. These are useful tools for data exploration and can help us separate thematic behaviors across sampled participants' game-plays Clustering allows us to group participants into distinct behavioral categories from their decisions throughout the simulation. Using these analyses, we can design experiments that can automatically group participants by their decisions, predict their behaviors, and even adapt the simulation based upon their group.

Several clustering algorithms have been validated for financial risk analysis [40] and have been applied to examine behavior in experimental games [41]. Comparing player strategies using a clustering framework can help identify appropriate audiences for tailoring interventions or personalized messaging. We selected the K-Means algorithm [42] to cluster the biosecurity adoption ratings and categorize observed behaviors. Mitigation strategies, recorded from participants' game-plays, spanned from minimally protective risk-tolerant behaviors to more cautious risk averse approaches.

Here we present a framework for digitally simulating risk scenarios to identify behavioral strategies from sampled participants. These digital experiments allow us to test the effects of various informational stimuli and their influence on the decision-making process. This can also help us understand in general how perceptions of risk and attitudes may differ across a sampled population. Our sampled audiences vary from industry professionals and stakeholders to the general public.

Our overarching goal is to simulate complex decision mechanisms using digital representations of disease outbreak situations. We analyzed participant choices to study behavioral strategies employed under different scenarios. To reach our goal, we designed two experiments to quantify behavioral risk profiles associated with a biosecurity investment response to outbreak scenarios.

Experiment One focused on identifying the most prominent behavioral strategies associated with risk mitigation in response to perceived economic danger. We designed a digital experiment [41] simulating the budgeting of a farm's biosecurity over the course of a year during an outbreak. We recruited individuals to participate in our experimental simulation using Amazon Mechanical Turk (Mturk), an online survey marketplace recently applied for behavioral experiments [43, 44]. In addition, specific treatments about visibility of infection and biosecurity in the producer network were implemented to test the following hypotheses:

(H1): More visibility in the number of infected sites increases risk aversion.

(H2): More visibility of the amount of biosecurity in the system increases risk taking behaviors.

In Experiment Two, we identify differences between our audience of primary interest, industry specialists, and the sampled population from Amazon MTurk. Here we sought to delineate behavioral differences between an audience with extensive knowledge of the swine industry and those without relevant industry experience (i.e., online recruits). We hypothesize that industry professionals will internalize [45] and empathize with diseased animals and/or past experiences during outbreak situations and thus behave differently than an audience without industry experience:

(H3): Industry professionals risk mitigation behaviors will differ from an audience without industry experience.

To test this hypothesis, we rented a booth at the 2018 World Pork Expo in order to recruit industry professionals and stakeholders to compare their decision-making strategies to an additional sample of online recruited participants without relevant industry experience. By examining behavioral differences between audiences, we may be able to determine how to leverage behaviors from an easily accessible population to gain insights that apply to a group that is logistically challenging to recruit.

## Materials and methods

Our digital experiments simulated the management of a porcine facility's biosecurity in the face of a contagious disease. Our simulation was modeled after [41]. Our updated version follows a similar user interface (UI) and mechanics, with the added capability for online deployment. Practices accepted by the University of Vermont Institutional Review Board were followed for experiments using human participants (University of Vermont IRB # CHRBSS-16-232-IRB). Instructional slideshows were presented to participants before they began play, and were identical for both experiments. The slideshows, describing the gaming mechanics and interface, are given in SI 1. We conducted two experiments.

### Experiment one

We recruited 1000 participants using Amazon Mechanical Turk, an online survey marketplace [46]. The experimental simulation application was built in the Unity Development platform and hosted with WebGL [33, 47]. In Fig 1, the simulation interface conveys information about each neighboring facility's biosecurity level as well as information about the facilities' disease infection status. We designed our digital experiment to simulate the management of swine production facilities. Players made management decisions to adapt their facility's biosecurity during several outbreak scenarios. We hosted our simulation online. Each participant played 32 rounds with each round consisting of up to 6 decisions. Each decision provided the opportunity to invest simulation funds towards biosecurity. One investment choice was allotted per

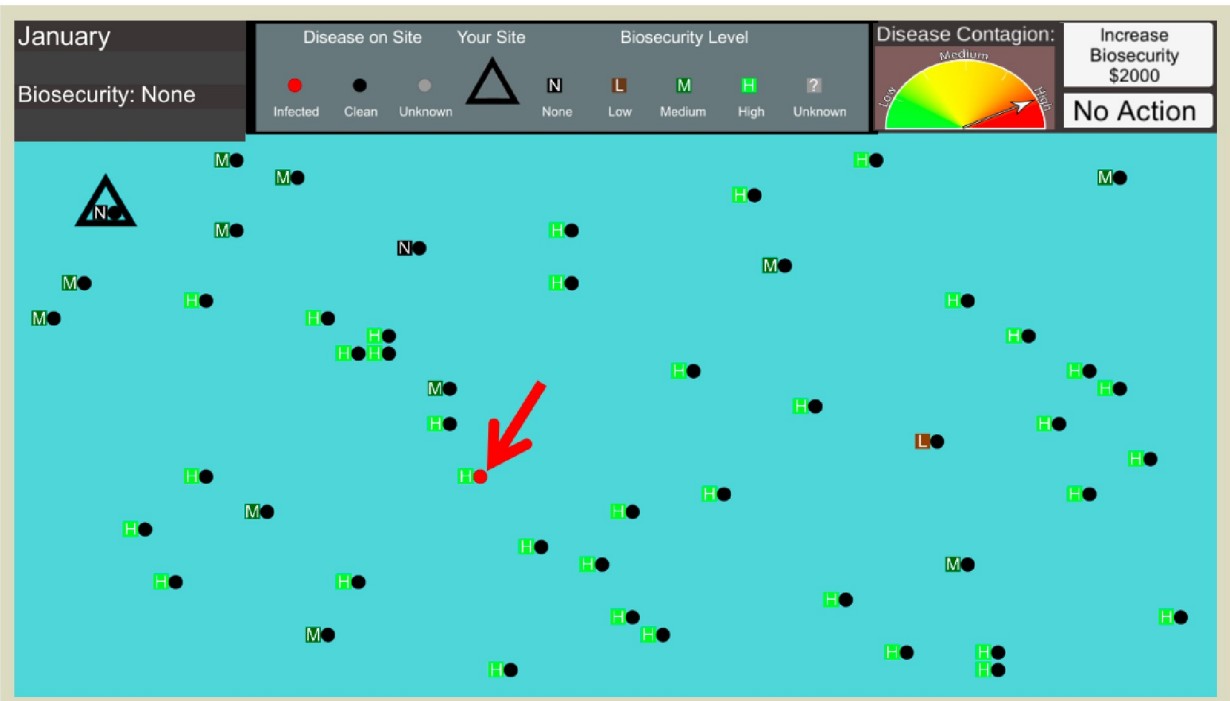

**Fig 1. User interface.** The red arrow marks an infected facility (red dot). The player's facility is enclosed by a triangle. Each round spans 6 decision months, where the player can remain at the current level of biosecurity or invest in increased biosecurity from None to Low, Medium, and High. An indicator for the infection status is presented to the participant.

month costing $1000 simulation dollars, which could be spent up to three times per simulated year: upgrading their status from "None to Low", "Low to Medium" or "Medium to High". The player was never economically impeded from investing in biosecurity (i.e., this option is always available independent from their total score). If the player owned facility became infected, the round would end and $25,000 was subtracted from their total score. At the end of the simulation, players were compensated with a rate of $50,000 in game currency to $1 USD.

The network of facilities in each round includes one player-controlled facility and 49 simulation-controlled facilities. Each computer-controlled facility in the simulation was assigned a biosecurity value. The probability of a contaminated facility infecting other premises was scaled with distance, facility-specific biosecurity and a predefined infection rate (Infection rates: Low (0.08) or High (0.3)). At the close of each decision month, the transmission probability per facility is determined using a pseudorandom number drawn from a uniform distribution.

Each round, consisting of up to six decision months, begins with a single infection. At the end of each month, information about the infection's progression is presented to the player along with the opportunity to invest experimental funds to increase their facility's biosecurity. Increasing biosecurity protects the player's facility by dampening the infection's transmission probability. The probability of infection must be inferred by the player using their information regarding the number of infections in the system, amount of biosecurity implemented at neighboring facilities, and the infection rate, which is clearly displayed on the user interface. Player installed biosecurity does not depreciate over time, nor are participants given the option to decrease their biosecurity level. If the player's facility becomes infected after a decision month,

the player is alerted, $25,000 experimental dollars are subtracted from their score, and the round ends immediately. The simulation then progresses to the next year with a new initial infection and spatial distribution of producers, and the player owned facility's biosecurity level is reset to "None".

Our simulation tested the effects of concealing information about infection status as well as the amount of biosecurity in the system at each of the computer-generated production facilities. In this way, we injected information uncertainty into the decision-making process, exposing each participant to a variety of risk scenarios. This allows us to test for differences in behavior given more or less information regarding infection status and/or the amount of biosecurity present at neighboring farms.

Participants played multiple rounds with treatments that varied the infection dynamics as well as the visibility of neighboring facilities' infection status and biosecurity level. In one quarter of all rounds, the number of infections and biosecurity status of each facility was fully visible to the participant. The remaining 75% injected uncertainty into the decision making process by concealing information regarding either the infection and/or biosecurity statuses across the production network. The neighboring facilities were all present on the user interface, however either the infection indicators and/or biosecurity statuses remain 'gray', indicating unknown. These treatments were incorporated to consider how the presence of uncertainty and increased risk of infection can affect players' decisions.

Each treatment was played twice, once with biosecurity values drawn from a Low biosecurity distribution and once with values drawn from a High biosecurity distribution. The Low biosecurity distribution generated biosecurity values for each facility drawn with 60% chance for 'None' rated at 0, a 32% chance of 'Low' (1), a 6% chance at 'Medium' (2), and a 2% chance at 'High' (3). The High distribution randomly pulled biosecurity values with a 60% chance for 'High' (3), a 32% chance of 'Medium' (2), a 6% chance at 'Low' (1), and a 2% chance at 'None' (0). Our treatments consisted of all permutations of Low, High biosecurity distributions against uncertainty in both infection information and amount of biosecurity in the system (i.e., 25% full information, 25% biosecurity obscured, 25% infection obscured, 25% no infection or biosecurity information). This accounted for 182,124 biosecurity investment decisions collected from the 1,000 participants. Our clustering analysis grouped each of these treatment scenarios in order to compare the overall comparative risk associated with each player's choices. We then grouped decision data during high and low information obscurity to test our behavioral hypotheses (H1,H2). The particular effects of these treatments on the decision making process were further explored in [41].

Each decision has an associated risk, based upon the amount of biosecurity implemented at the player's facility and severity of the infection rate. The infection spreads to more facilities per month, each of which can infect the player's facility. Biosecurity investments reduce infection rates for the entire round, hence, earlier adoption of biosecurity leads to reduced risk throughout the six month round. Players can invest their simulated earnings to reduce their risk of infection, or take a chance with a lower disease protection and a possibly higher end-round payout.

Each participant's decisions were assigned biosecurity adoption ratings depending on the amount of biosecurity they implemented during the simulation. More risk averse strategies choose to increase biosecurity earlier within the simulated year, while risk tolerant strategies allocate fewer experimental dollars to biosecurity in a gamble for a higher payout. Every individual, $i$, was assigned a biosecurity adoption rating, $R_i$, computed using decisions, $d \in D_i$, from each simulated month. Each participant's risk score is calculated by tallying the player facility's level of biosecurity, $b_d \in \{0 = \text{"None"}, 1 = \text{"Low"}, 2 = \text{"Medium"}, 3 = \text{"High"}\}$ across

each simulated year and then normalizing by the total number of player decisions, $|D_i|$:

$$R_i = \frac{1}{|D_i|} \sum_{d \in D_i} b_d \tag{1}$$

for each decision, $d$, of the $i^{th}$ player. The biosecurity adoption rating, $R_i$, increases as the player invests in more biosecurity, which decreases their risk of infection. This means that investments in biosecurity that occur in earlier months carry more weight, because early investments increase a facility's protection for the remainder of the round (i.e., each decision month). The most risk averse strategy is characterized by those participants that consistently invested substantial funds into biosecurity during the early months of each round.

We clustered these strategies using a K-means clustering algorithm [42] implemented with the Python programming language [48]. Graphics were created using matplotlib [49]. Here, the clustering coefficient, $K$, is the number of unique clusters assumed for each analysis. We chose $K = 3$, using the elbow method (Fig 2), as it optimizes the sum of the square errors across cluster centers [50, 51] and produces the most pronounced scheme of observed behavior.

The perceived risk attitudes of each recruited participant were calculated using each of their investment decisions. Each player's two dimensional biosecurity adoption rating was calculated from each of their investment decisions aggregated from two datasets: (1) decisions made given rounds with Low infection rates, and (2) decisions made in rounds with High infection rates, $\vec{R}_i = (R_{i(Low)}, R_{i(High)})$. We tested the effects of restricting information regarding the presence of infection as well the visibility of biosecurity statuses of neighboring facilities. This injected uncertainty into the decision making process. Using the biosecurity adoption ratings as a measure of risk, we test for a significant difference in the distributions using one-tailed Mann-Whitney U tests [52]. This non-parametric test was chosen since each distribution of biosecurity adoption ratings failed D'Agostino and Pearson's test for normality [53, 54].

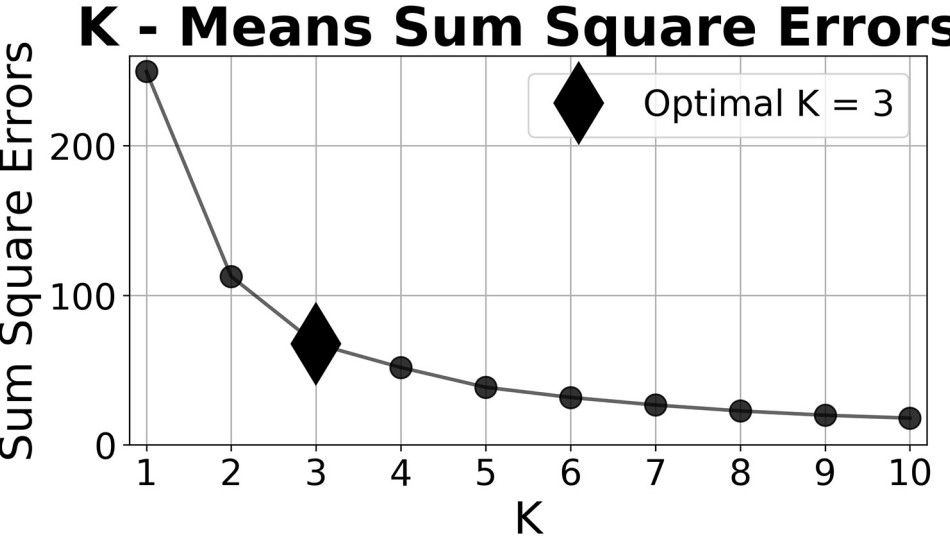

**Fig 2. Risk K means clustering.** For each clustering coefficient, $K$, the sum of the squared errors are plotted using participant's Biosecurity Adoption Ratings, $\vec{R}_i$.

## 0.1 Experiment 2

To test for a difference in strategy by audience, we directly compared decisions of two groups: Amazon Mechanical Turk and industry professional participants. For this analysis, we used a Biosecurity Investment version of our experimental simulation that featured a single infection rate across 32 rounds (i.e., simulated infection scenarios).

We hosted a booth at the 2018 American World Pork Expo [55] and recruited 50 participants with knowledge of the swine industry including business owners, production managers, laborers, animal health experts and enthusiasts. In contrast, this simulation featured a constant infection rate (0.15), an intermediate value between the low (0.08) and high (0.3) infection rates tested in Experiment One. This allowed us to collect sufficient observations for a statistical comparison, while decreasing the total participation time, which aided in our enrollment. We recruited an additional 50 Amazon Mechanical Turk participants to participate in this experiment. Each audience type played 32 rounds with 6 decision months per round, thus providing 9,600 decisions per cohort. Decisions by each group were compared using a two sample Kolmogorov-Smirnov (KS) test [56]. At the end of the simulation, Mturk recruits were compensated with a rate of $23,500 in simulation currency to $1 USD. We paid a higher rate for participants at the World Pork Expo in order to bolster participation: $12,000 simulation dollars to $1 USD.

# Results

## 0.2 Experiment one

In Fig 3, we clustered each participants biosecurity adoption ratings using K-means clustering with $K = 3$. The circles represent each player's two dimensional risk attitude rating, $\vec{R_i} = (R_{i(Low)}, R_{i(High)})$. The diamonds portray the center of each cluster. The x axis scores their decisions using a low infection rate (0.08) while the y axis represents scores from a high infection rate (0.3). Near the origin, (0,0) players adopted very little biosecurity during the experiment. In the upper right corner, players adopted the most biosecurity for both infection rates. Each point (or cluster) shows how differently a player's decisions behaved in the two treatments. Points close to the main diagonal (dotted black line along $y = x$) do not modify their behavior in response to the game context (e.g., always or never investing in increased biosecurity), while those points off the main diagonal show players who differ their behavior in response to opportunities in game situations. The bottom right quadrant is empty, as these scores represent nonsensical behavior (i.e., only adopting high biosecurity on low risk rounds and low biosecurity on high risk rounds).

Cluster 1 (♦) made the most *risk averse* biosecurity investment decisions. They adopted the most biosecurity, for both low and high infection rates, in comparison to the other clusters. Cluster 2 (♦) took the opposite approach, adopting the least amount of biosecurity in both dimensions. These *risk tolerant* participants attempted to maximize their payouts using a minimal biosecurity investment strategy.

Cluster 3 (♦), the *opportunists*, adopted more biosecurity with a high infection rate and little to no biosecurity with the low rate. Some cautious members of this group purchased more biosecurity than the risk averse group (Cluster 1) during highly contagious rounds. This group of players is characterized by a balance between risky behavior when the probability of transmission was dampened and more conservative choices when presented with a higher risk of infection. They behave similarly to the risk-tolerant during low infection rates, and appear more risk averse during highly infectious rounds.

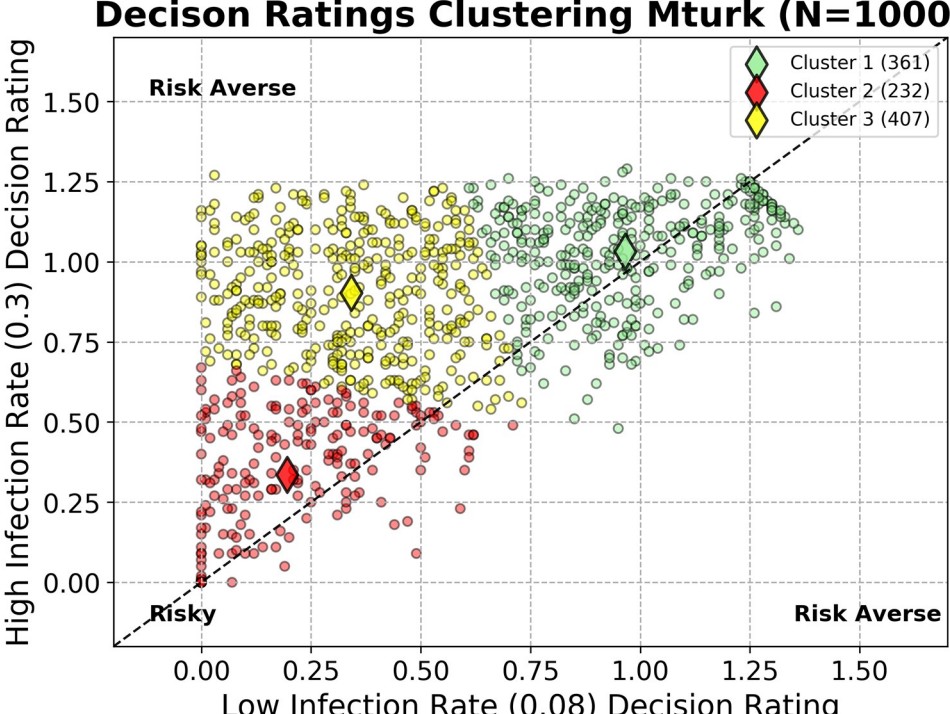

**Fig 3. Risk cluster analysis.** Participant biosecurity adoption ratings are clustered using the K-means algorithm with $K = 3$. The circles represent each player's two dimensional risk attitude rating, $\vec{R_i} = (R_{i(Low)}, R_{i(High)})$. The diamonds portray the center of each cluster. The x axis scores their decisions using a low infection rate (0.08) while the y axis represents scores from a high infection rate (0.3). Near the origin, (0,0) players adopted very little biosecurity during their game-play. In the upper right corner, players adopted the most biosecurity for both infection rates. Points close to the main diagonal do not modify their behavior in response to the game context, while points off the main diagonal show players who differ their behavior in response to simulated opportunities.

For reference, the optimal or *risk neutral* strategies can be quantified from the end of round earnings ($E = \$15,000$), probability of infection ($p_i$), cost associated with becoming infected ($C_i = \$25,000$) and cost of biosecurity ($b_c = \$1,000$ per level). We ran several hundred trials given each level of biosecurity to estimate the probability of infection given each biosecurity level. The expected return ($E$) for each biosecurity adoption choice was calculated in experimental dollars as: $E = (E - b_c) * (1 - p_i) - (p_i * C_i)$. In Table 1 we show the expected return for biosecurity adoption choice with respect to infection rate. We see that a minimum biosecurity status of 'Low' during Low infection rates and a "High" biosecurity status during High Infection rates are the most optimal choices for producing the highest expected returns.

**Table 1. Biosecurity level expected returns.** Expected Returns in experimental dollars for each level of biosecurity adopted. The probability of infection ($p_i$) is estimated using several hundred trials at each specified biosecurity level. Recall, the the probability of infection depends on the infection rate and distance to each infected facility.

| Biosecurity Level | Expected Earnings Low Infection Rate ($p_i$) | Expected Earnings High Infection Rate ($p_i$) |
|:---:|:---:|:---:|
| None | $12,204.30 (7%) | -$1,729.22 (41.8%) |
| Low | $12,357.89 (4.2%) | -$2,044.30 (41.1%) |
| Medium | $11,400.00 (4.2%) | $400.00 (33.2%) |
| High | $11,406.42 (1.6%) | $4,857.91 (19.3%) |

In Fig 4 we highlight the differences in these observed behaviors with histograms of each cluster's decisions to invest in biosecurity as a function of decision month. Here, the *risk tolerant* adopt little biosecurity for both low and high infection rates, while *risk averse* players tend to frequently increase protection. The *opportunists* mirror risk-tolerant behavior when the infection rate is low and appear more risk averse when infection rate was presented as high. We can show this by comparing distributions by computing their Kullback–Leibler (KL) divergence [57]. When comparing the monthly distributions of biosecurity investment decisions, *[No Biosecurity, Low, Medium, High]*, from Opportunists (OP) versus Risk Tolerant (RT) during low infection rates, we found:

$$D_{KL}(OP\|RT) = [0.0002, 0.052, 0.0482, 0.045]$$

Similarly, when comparing Opportunists (OP) to the Risk Averse (RA) group during high infection rates we find:

$$D_{KL}(OP\|RA) = [0.0005, 0.04, 0.0351, 0.0186]$$

This helps justify our intuition regarding the similarities between these groups under these conditions.

To test hypotheses (H1,H2), we first investigated how the risk cluster distributions, {Risk Averse (RA), Risk Tolerant (RT), Opportunistic (O)}, may change with respect to each set of information visibility treatments. We calculated each participant's two-dimensional risk scores, ($R_{i(Low)}$, $R_{i(High)}$), for each group of visibility treatments and then re-categorized each participant's biosecurity adoption rating using the centroids defined from the full decision space (see Fig 3). We then applied a KS test to compare the differences in behavioral groups between treatments.

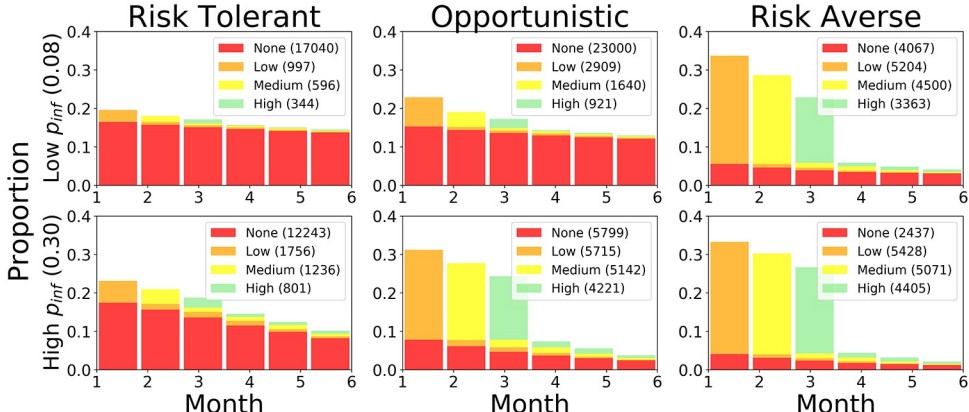

**Fig 4. Cluster comparison.** Histograms of the proportions of all decisions to remain with no biosecurity ("None") or increase from None—to Low—Medium—and finally to High as a function of decision month. Biosecurity can only increase one level per month. Less biosecurity was implemented when the infection rate, $p_{inf}$ was Low (= 0:08); the number of decisions to invest in biosecurity increased with higher infection rates. The Risk tolerant cluster (left column) implements the least biosecurity, while the Risk Averse cluster (right column) invests the most biosecurity under both infection rates. After attaining a High biosecurity level, no more decisions can be logged for the simulated year, which is why most of the decisions from the risk averse cluster are completed by the third decision month. The Opportunistic cluster (middle column) behave like the risk averse group under high infection rate scenarios, and implement less biosecurity during low infection rates.

For infection visibility information treatments (H1) the change in clustered distributions was not significant (KS $D = 0.045$, $p = 0.257$, two tailed). However, we did find more risk averse behaviors during full infection visibility treatments (395 RA, 207 RT, 398 O) compared to low infection visibility (350 RA, 290 RT, 360 O). See S2 Fig for additional graphs showing the clustered risk scores per visibility treatment.

For biosecurity information treatments (H2), we found a significant difference (KS $D = 0.08$, $p < 0.005$, two-tailed) in the clustered distributions when comparing treatments with full visibility of neighboring biosecurity (322 RA, 290 RT, 388 O) to low biosecurity visibility (402 RA, 207 RT, 391 O). This lends support to (H2), as we see more risk taking behaviors with more visibility of biosecurity in the system. This helps justify our intuition regarding the similarities between these groups under these conditions.

In order to formally investigate hypotheses (H1) and (H2), we computed each participant's aggregate biosecurity adoption rating across both the low and high infection rates for each set of information treatments. To test (H1), we examined whether more visibility in the number of infected sites increased risk aversion. To accomplish this, we measure each participant's biosecurity adoption rating during 16 treatments in which the infection status of neighboring facilities was visible {median = 1.40, $\mu = 1.40$, $\sigma = 0.62$, min = 0, max = 2.50}, versus 16 treatments with hidden infection statuses {median = 1.31, $\mu = 1.30$, $\sigma = 0.69$, min = 0, max = 2.50}. Using the Mann-Whitney U Test, we found the distributions of biosecurity adoption ratings differed significantly, with more biosecurity being implemented when the infection status was fully visible (Mann–Whitney $U = 541840.5$, $n_1 = n_2 = 1000$, $p < 0.001$, one-tailed).

For (H2) we tested if more visibility of the amount of biosecurity in the system increases risk taking behaviors. We similarly compare 16 treatments in which biosecurity was visible {median = 1.27, $\mu = 1.27$, $\sigma = 0.63$, min = 0, max = 2.50} versus 16 treatments in which the neighboring biosecurity statuses were hidden {median = 1.45, $\mu = 1.43$, $\sigma = 0.66$, min = 0, max = 2.50}. This difference in biosecurity adoption ratings between distributions was significant, with less biosecurity being implemented during treatments in which neighboring biosecurity statuses were visible (Mann–Whitney $U = 429424.0$, $n_1 = n_2 = 1000$, $p < 0.001$, one-tailed).

## 0.3 Experiment two

To test (H3), compared the decisions made by industry professionals recruited at the 2018 World Pork Expo to the decisions made by workers from Amazon Mechanical Turk. In Fig 5, the risk aversion distributions are given for both the Amazon Mechanical Turk {$\mu = 1.41$, $\sigma = 0.71$, median = 1.34, min = 0.02, max = 2.50} and World Pork Expo {$\mu = 1.36$, $\sigma = 0.66$, median = 1.31, min = 0.13, max = 2.50} audiences. Using a two sample KS test, we found ($D = 0.16$, $p = 0.51$, $n_1 = n_2 = 50$), leading us to fail to reject the null hypothesis that the two distributions of two samples are the same. Results did not detect a difference in the spectrum of behavioral strategies from sampled online participants and agricultural professionals under our risk aversion metric. For comparison, we also find the same result using a two tailed Man Whitney U Test ($U = 1180.0$, $p = 0.63$, two-tailed, $n_1 = n_2 = 50$).

## Discussion

In this study and in other published works [20, 35, 41] we have demonstrated the potential value of using experimental gaming simulations as a data gathering tool for use in categorizing behavioral strategies. Our experimental framework focuses on creating a digital representation of a complex decision mechanism in order to identify prevalent behavioral strategies.

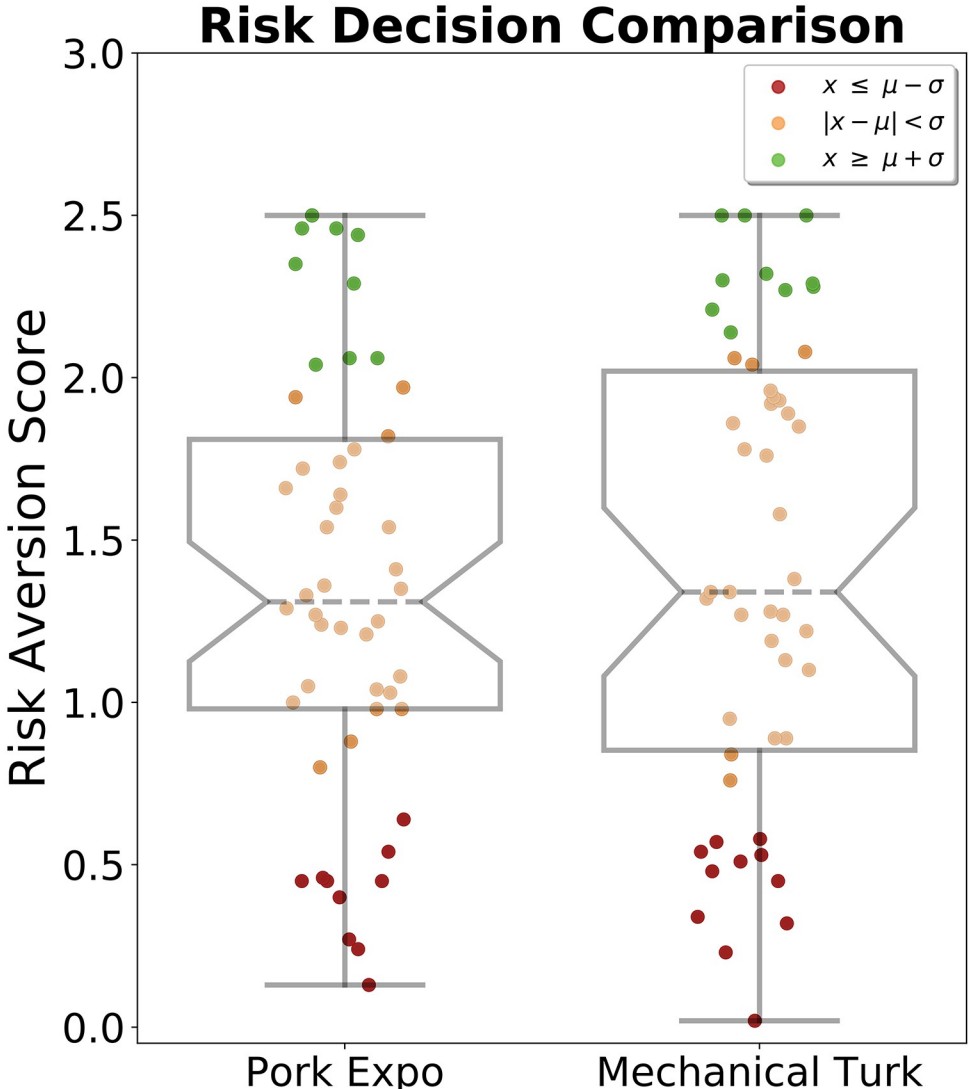

**Fig 5. Mturk—Industry comparison.** Biosecurity Adoption Ratings from 50 industry professionals who attended the 2018 World Pork Expo were compared to 50 participants recruited online from Amazon Mechanical Turk. Each participant's rating ($R_i$) was calculated from a single infection rate (0.15) across 192 decision months for a total of 19,200 choices.

Experimental gaming simulations are readily applicable for designing controlled settings for collecting assessments of tasks in response to changing visual stimuli [35]. In Experiment One, we created a personalized metric for comparing each player's risk mitigation decision strategy and identified three distinct behaviors. We tested how risk communication and information accessibility of both the disease infection status and neighboring biosecurity protection efforts affect this decision-making process.

Gaming simulations for education, or edutainment are more dominant for cognitive gain outcomes as contrasted with traditional learning methods [58]. Our framework may also be adapted to create educational tools featuring risk messaging and communication. Feedback on the consequences of the decision making process can help a player learn from their past decisions. We can also use these educational tools for data gathering and testing the effects of

presenting risk communication strategies with respect to a complex decision mechanism. Since these applications can be hosted online, they can allow us to collect inputs from a wide audience for computational social science research applications, while also providing educational value to the players.

We tested the effects of information uncertainty with regards to the infection and biosecurity status of neighboring farms. In (H1) we found more visibility in the number of infected sites results in an increase in the biosecurity adoption rating distribution (i.e., more risk averse behaviors). This is intuitive as there is more perceived risk when the participant can see the spread of the disease over the course of the year. This feedback appears to invoke more risk averse behaviors in our sample. In (H2) we found more visibility in the amount biosecurity implemented throughout the system increases risk tolerant behaviors. This could be due in part to the free-rider effect in which some participants venture for a higher payout by exploiting their neighboring biosecurity status, perceived as a shield from the infection.

While Amazon Mechanical Turk provided a tool for fast recruitment of many participants for our experiment, we assume that, based on the overall rate of employment in agriculture, most online participants were not currently working in an agricultural field. Therefore, their decisions may potentially differ from experienced industry professionals. Experiment Two focused on comparing the decisions from industry professionals and stakeholders to a random sample of participants recruited from Amazon Mechanical Turk. We did not find evidence to support (H3). No difference was detected in the proportion of observed risk scores of the two cohorts. This may be in part due to our relatively small sample size, approximately 20,000 decisions spanning 100 participants, which was in part due to the difficulty in obtaining decision-making data from industry professionals. Through survey methods, [59] found farm owners self-identified as generally more risk tolerant in comparison to the general public, however were more risk averse in comparison to non-farm business owners. Our simulation was framed specifically for outbreak mitigation and resource allocation, which may have influenced the risk tolerance of farm owners due to the internalization of the economic consequences of their decisions.

Naturally, we may expect some differences in how an experienced industry professional approaches these simulated risk scenarios, given any real-world past experiences mitigating their disease risk. However this data, suggest that the underlying behavioral distributions are comparable under our risk aversion metric. Since experimental data from industry professionals is difficult and costly to gather at scale, these results lend credence to behavioral results from data collected using a convenience sample of online recruited participants for analyzing behavior even if the participants lack insider knowledge of the industry being studied.

Effective risk communication strategies are essential for crisis aversion and mitigation [60]. In particular, outreach messaging strategies may need constant adaptations in order to improve compliance at critical moments to minimizing outbreaks [61]. Online recruitment of participants can be used to rapidly gather data for testing the efficacy of risk messaging strategies. Comparing their decision strategies to industry professionals and stakeholders helps leverage our findings. This framework can help us study behavioral mechanisms leading to more proactive risk management. These interventions can then be further tested using simulation modeling, such as agent-based modeling approaches to forecast their effect on systemic contagion dynamics [20]. Additionally, this framework can be leveraged to study risk aversion with respect to other behavioral strategies or to investigate the sensitivity of behavioral responses and how they change over time (i.e., learning effects).

Implications of these results for the industry itself include the need to appreciate the heterogeneity along the risk aversion and risk tolerance spectrum that is apparent, not just in Amazon Mechanical Turk participants, but industry professionals. Risk communication and

incentive strategies likely need to be tailored for specific populations of industry producers. In our study, we focused on analyzing risk aversion with respect to biosecurity investment and disease prevention, since this objective decision making process can impact the wellbeing of these agricultural systems. Those producers who tend to be more risk averse, in general, may only need basic information regarding risks of disease and consequences to be incentivized into action. Other, more risk tolerant populations, may require mandates to motivate change. Those producers who take a more situational approach, who essentially learn as they interpret changes in standard operating procedure over time, may benefit from extended learning and training opportunities. In future simulations, the use of incentives to nudge participants toward higher levels of biosecurity investment should be tested as a potential intervention for this population. Both the risk tolerant and the risk averse populations were reluctant to change strategies and may require larger incentives (higher penalties for a disease strike or higher monetary incentives to adopt biosecurity) or the use of regulatory power to improve resistance to disease incursion and ensure system resilience.

## Conclusion

Experimental gaming simulations can provide insights into a wide variety of social ecological systems and further, data collected can be used to provide inputs to digital decision support systems. Interfaces can be adapted for a wide variety of applications. Recruitment of participants can be rapidly assembled using web hosted platforms. Tailored interfaces that simulate real-world decisions can control information variability to capture the choices of individuals.

Our digital experiments can improve upon traditional survey methods using immersive simulations for tracking human behavior. Online survey recruitment tools, like Amazon Mechanical Turk, can assist in expediting recruitment for conducting digital experiments. Our results validate using online participants for accelerating behavioral studies. We affirm this by comparing behaviors to a convenience sample of industry professionals, which can be more difficult to recruit. Further validation should be conducted to compare results from industry professionals to online recruited audiences.

Applying a clustering algorithm to our risk aversion metric (Eq 1) helped us identify the most prominent behaviors exhibited by our sampled participants. We uncovered three distinct strategic clusters using a risk aversion scale to categorize participant's decision-making behavior. *Risk averse* participants invested the most resources to protect their facility, regardless of the infection rate. *Risk tolerant* players would invest little to nothing in biosecurity regardless of the communicated infection rate. *Opportunists* would take their chances with little protection when the infection rate was low, but invested in high protection when their perceived risk of infection increased. The opportunists were the most responsive to information provided. Identifying these types of behaviors may be beneficial for more targeted information campaigns in order to promote more resilient and healthier systems [20].

Our categorization of risk tolerant, opportunistic and risk averse strategies can help us model agricultural decisions and the ramifications of interventions that seek to alter behavior. Experimental gaming simulations can be applied to test the efficacy of risk communication information campaigns. Incentives can be incorporated into our simulations to test the effects of *nudging* [62] populations towards healthier risk management practices. Identifying realistic decision response distributions from tested human behaviors can help modeling approaches find system-wide optimal biosecurity resource allocation for outbreak mitigation.

Behavioral clustering has value because it allows us to identify a wide spectrum of behaviors and consider targeted interventions to groups who are more responsive to risk communication. People in both the risk averse and risk tolerant clusters, which make up 59.3% of the

tested population, do not appear to readily change behavior with additional information. These categories show recalcitrant behavior, and may be indicative of individuals that are "set in their ways". These populations are likely the most difficult populations to nudge towards alternate, and more productive pathways. Behavioral interventions of these populations will likely require substantial effort with associated costs and may not provide movement towards the desired outcomes. Yet changing communication strategy by altering disease and biosecurity communication strategies (H1 and H2) confirms that shifts in behavior are possible. We suggest that interventions may be best targeted towards those identified as risk opportunists because they are likely to change their behavior as information is modified. Further work can examine if messaging may differentially effect specific clusters, and thus, provide support for nuanced behavioral nudges designed to impact specific subsets of society.

## Supporting information

**S1 Data. The instructional slide show of "Protocol Adoption" gaming simulations presented to each participant.** A demo of the game is hosted online at: (https://segs.w3.uvm.edu/demos/protocol).
(ZIP)

**S1 Fig.**
(PDF)

**S2 Fig. Visibility clustering analysis.** Participant biosecurity adoption ratings are clustered for each set of visibility treatments. Cluster 1 are Risk Averse (green), Cluster 2 are Risk Tolerant (red) and Cluster 3 are Opportunistic (yellow). The top row compares Low neighboring biosecurity visibility (i.e., high uncertainty) to high biosecurity visibility. The bottom row compares infection visibility treatments. We found a significant difference in the clustered risk distributions for biosecurity visibility treatments. Histograms explicitly show the cluster differences between information uncertainty treatments.
(PDF)

## Acknowledgments

The authors would like to thank Susan Moegenburg, Ph.D., for assisting in project framing, data collection, and managerial support for this research project.

## Author Contributions

**Conceptualization:** Eric M. Clark, Scott C. Merrill, Asim Zia.

**Data curation:** Eric M. Clark.

**Formal analysis:** Eric M. Clark.

**Funding acquisition:** Christopher Koliba.

**Investigation:** Scott C. Merrill, Julia M. Smith.

**Methodology:** Scott C. Merrill.

**Project administration:** Scott C. Merrill, Christopher Koliba, Julia M. Smith.

**Software:** Eric M. Clark, Luke Trinity.

**Supervision:** Scott C. Merrill.

**Visualization:** Eric M. Clark.

**Writing – original draft:** Eric M. Clark.

**Writing – review & editing:** Scott C. Merrill, Luke Trinity, Gabriela Bucini, Nicholas Cheney, Ollin Langle-Chimal, Trisha Shrum, Christopher Koliba, Asim Zia, Julia M. Smith.

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
