## [Decision Letter · Decision Letter 0]

29 Jan 2020

Using Experimental Gaming Simulations To Elicit Risk Mitigation Behavioral Strategies For Agricultural Disease Management

PONE-D-19-29852

Dear Dr. Clark,

We are pleased to inform you that your manuscript has been judged scientifically suitable for publication and will be formally accepted for publication once it complies with all outstanding technical requirements.

With kind regards,

Judi Hewitt

Academic Editor

PLOS ONE

1. Please provide additional details regarding consent from the American World Pork Expo participants. In the ethics statement in the Methods and online submission information, please ensure that you have specified (a) whether consent was informed and (b) what type you obtained (for instance, written or verbal, and if verbal, how it was documented and witnessed). If the need for consent was waived by the ethics committee, please include this information.

Reviewers' comments:

Reviewer's Responses to Questions

**Comments to the Author**

1. Is the manuscript technically sound, and do the data support the conclusions?

Reviewer #1: Yes

2. Has the statistical analysis been performed appropriately and rigorously? 

Reviewer #1: I Don't Know

3. Have the authors made all data underlying the findings in their manuscript fully available?

Reviewer #1: Yes

4. Is the manuscript presented in an intelligible fashion and written in standard English?

Reviewer #1: Yes

5. Review Comments to the Author

Reviewer #1: I enjoyed reading this paper, it is clear and interesting. it will provide a good contribution to understanding how people respond to perceived risk and would be of interest to a wide audience, particularly in terms of the methods used.

However it is beyond my expertise to review the statistical component.

One suggestion is to expand the thinking from using the new knowledge for communication of risk future interventions to using the game simulations to support learning in play. It would also be useful to record the learning post game, and record any difference in action between any repeat plays. In short, games can be more than just data collection they can be used to create reflections and learning within the test group. Do you have any data from your experiment on this aspect? are there any repeat play?

6. PLOS authors have the option to publish the peer review history of their article (what does this mean?). If published, this will include your full peer review and any attached files.

Reviewer #1: No

---

## [Editor Report · Acceptance letter]

26 Feb 2020

PONE-D-19-29852 

Using Experimental Gaming Simulations To Elicit Risk Mitigation Behavioral Strategies For Agricultural Disease Management 

Dear Dr. Clark:

I am pleased to inform you that your manuscript has been deemed suitable for publication in PLOS ONE. Congratulations! Your manuscript is now with our production department. 

With kind regards,

on behalf of

Dr. Judi Hewitt 

Academic Editor

PLOS ONE